# Increasing young people's use of sexual and reproductive health services in government health facilities in rural Kenya

Paula Tavrow[1]*, Krupa Maria Varghese[1], Albert Obbuyi[2¤a], Collins Juma[2¤b]

**1** Department of Community Health Sciences, Fielding School of Public Health, University of California, Los Angeles, California, United States of America, **2** Centre for the Study of Adolescence, Nairobi, Kenya

¤a Current address: Safari Doctors, Lamu, Kenya
¤b Current address: Operations Department, Tiko Africa, Bungoma County, Kenya
* ptavrow@ucla.edu

## Abstract

Young people in sub-Saharan Africa, especially in rural areas, face numerous daunting geographic and psychosocial barriers to accessing sexual and reproductive health (SRH) services and information. Adolescents worry that providers will be judgmental and not accord them privacy and confidentiality. They are also concerned that community members or extended family may see them waiting for SRH services, which could lead to parental perceptions of promiscuity and punishment. Because they lack comprehensive sexuality education (CSE) in schools, youths often have serious misconceptions about contraceptives. To address these barriers and increase youths' uptake of SRH services, the After-Hours Adolescent Project (AHAP) was created as a low-cost intervention for rural government health facilities. The main strategies involved extending clinic hours into early evening to be more convenient and confidential for youths, and training newly-graduated nurses to serve as AHAP nurses. AHAP was tested using a randomized cluster design in thirteen government health facilities in Western Kenya. The study examined two intervention types—the full intervention, which included having AHAP nurses conduct CSE classes in nearby schools and community locations; and a partial intervention, which was entirely clinic-based. After one year, AHAP increased young people's use of SRH services by 87% (full intervention 97%; partial intervention 77%), whereas comparison facilities saw no change. The full intervention facilities also attracted significantly more female and younger SRH clients. Additionally, improved nurse SRH attitudes after AHAP training were sustained. In sum, AHAP was acceptable and effective in rural settings, but requires ongoing budget outlays for nurse salaries.

## Introduction

It is estimated that young people aged 10–24 constitute close to a quarter of the world's population.[1] Among the world regions, sub-Saharan Africa has the largest

**Data availability statement:** All relevant data are within the manuscript and its Supporting information files.

**Funding:** This study was funded by grants from the Lalor Foundation (#20182859) and Save the Children (Sweden). The funders did not have any input into the conduct and analysis of the study, the writing of the manuscript or the decision to submit the manuscript for publication. There was no additional external funding received for this study.

**Competing interests:** The authors have declared that they have no competing interests.

percentage of young people, with about half of the population below the age of 18.[2] It also has significant sexual and reproductive health (SRH) challenges, particularly among adolescents.[3,4] The region accounts for the highest HIV burden globally, with HIV/AIDS being a leading cause of adolescent mortality, yet only about 25% of adolescents have ever been tested.[3,5] Sub-Saharan Africa also has the highest rate of adolescent pregnancies, with over 200 births per 1000 girls.[3] About 45% of adolescent pregnancies in the region are unintended, and less than half of African girls report that their family planning needs were met.[6,7]

Like many other African countries, Kenya's young people have high levels of sexually transmitted diseases and unwanted childbearing. Early sexual debut is common, with 47% of women and 55% of men in Kenya reporting initiation of sexual intercourse before age 18.[8] More than half of all new HIV infections occur among adolescents and young adults.[8] With limited access to contraceptive services and legal abortions, Kenya's teenage pregnancy rate is 15%, which is associated with risky, unsafe abortions, pre-eclampsia, birth complications, depression, and school dropouts.[9] In rural areas, girls' pregnancy rates are higher, and they attempt to abort later, when it is more dangerous.[10] Only 10% of Kenyan students who give birth manage to return to school.[11] This contributes to a vicious cycle of poverty and high fertility, since early childbearing is associated with larger families, lower use of contraception, and fewer career options.[12]

## Barriers to adolescent access to information and services

Historically, sub-Saharan Africa has lagged in offering comprehensive SRH education and services to adolescents.[13] A dearth of comprehensive sexuality education in schools means that young people generally lack accurate information about SRH and are often unaware that they can obtain family planning counseling and methods in health facilities without parental permission.[14,15] Providers' conservative values and attitudes may also limit the SRH information they give to young people. Moreover, internet access in rural areas is limited, thereby preventing young people from accessing SRH information on their own. Societal taboos about discussing sex inhibit conversations between adolescents and parents, teachers, or clinicians.[14,16] Rural parents report feeling unequipped to discuss sexual health with their children.[17]

Concerning SRH services, many young people in Kenya and other African countries lack convenient access to health facilities, particularly in rural areas. Government health facilities have frequent contraceptive stock-outs, long waiting times, limited hours, and a lack of privacy and confidentiality.[14,18] Furthermore, conservative social norms and negative provider attitudes and behavior can deter young people from seeking SRH services.[4] Although Kenya's National Guidelines for Provision of Adolescent and Youth-Friendly Services emphasize the importance of catering to the specific needs of adolescents and young people, few health providers in rural areas have been trained to provide youth-friendly services.[19] Health providers sometimes refuse to offer contraceptives to unmarried or nulliparous adolescents due to beliefs that adolescents should be abstinent until marriage, that condoms will cause promiscuity, or that hormonal contraceptives will lead to infertility or other harms.[8,20,21]

Adolescents frequently report that providers are judgmental, uncaring, and disrespectful.[14,20] A recent study conducted in Kenya, South Africa, and Zimbabwe found that healthcare providers' negative views about adolescent sexual activity could pose a major obstacle to girls' obtaining oral pre-exposure prophylaxis.[22] Fear of receiving discouraging advice or judgment from providers and general concerns about unfriendliness have hindered youth uptake of SRH services.[23]

Relatedly, the older age of providers seems to impede youths' accessing of services. Adolescents in Kenya have reported feeling uncomfortable sharing their concerns with older adults.[14,23] One study found that young providers have a positive effect on SRH service uptake.[23] Providers themselves sometimes report that young clients are troublesome and suggest that these clients may feel more at ease with younger health workers.[22] Older providers describe language barriers because they do not understand the slang used by youths seeking SRH services. Younger providers who can connect with their clients and provide services in a friendly and empathetic way, rather than as parental figures, are generally believed to be more effective.[19]

Young people are also hesitant to access SRH services if they believe that their privacy and confidentiality will not be respected.[8,20,24] They fear being seen obtaining condoms or contraceptives due to social stigma and parental disapproval. They also worry that providers, especially in rural areas, will pass on information about them to others.[14] Long waiting times and having to wait outdoors or in specified SRH areas also undermine confidentiality. Young people fear being spotted by relatives or acquaintances of their parents. Those with means sometimes travel to facilities further away to avoid being seen by those in their community.[8]

Health facilities' operating hours and distance pose other major challenges to young people's SRH use. Not only are service times inconvenient, but service providers are not always present during official hours, or they close clinics early.[14,23] When health facilities are not open after school or on weekends, students are put in a difficult position of having to disclose health information to teachers to get permission to be excused from class.[8] In a study in South Africa, adolescents reported being unable to access services because they were in school during clinic hours and were harassed by providers if they came late to a clinic.[25] Moreover, young people may have difficulty reaching facilities or affording SRH services. Adolescents in rural areas usually lack funds for transport, consultation fees, pregnancy or STI tests, and supplies. Unless SRH services are free or very low-cost, adolescents will have trouble accessing them.[8]

As a result of the barriers to obtaining SRH services at public health facilities, pharmacies have emerged as an alternative source of contraceptives.[26] Young people have reported preferring pharmacies over health facilities because they perceive them as more convenient and private. [23] They considered pharmacy personnel to be less judgmental compared to facility staff. Additionally, when pharmacy personnel were similar in age to adolescents, they felt more comfortable interacting with them.[27,28] However, pharmacies have been shown to provide inadequate and sometimes inaccurate counselling on contraceptive use to adolescents.[26,28] Pharmacies also have a limited range of contraceptives and may not be affordable.[26] To reach all young people effectively with comprehensive SRH care and contraceptive services, it therefore continues to be important to increase access to adolescent-friendly government health facilities.

## Youth-friendly service delivery standards and framework

About a decade ago, the WHO developed global standards for adolescent SRH that called on health facilities to improve adolescent health literacy, encourage community support for the utilization of services, offer an appropriate package of services, have competent and non-judgmental providers, be open during convenient hours with required infrastructure and supplies, offer services without discrimination, collect and use data, and involve adolescents in decision-making.[29] However, systematic reviews of interventions to improve SRH services for adolescents show that few have incorporated most or all of these standards.[12,30] One review noted that only interventions that integrate both demand-side and supply-side activities could be considered ready for scaling up. Most interventions focused strictly on training staff or facility improvements, which did not have major impact.[31] In addition to gaps in intervention design, reviews have also called for rigorous evaluations of interventions to address the weak evidence base on effectiveness.[30,31]

To delineate the supply and demand-side factors that influence young people's use of SRH services, a conceptual framework had been developed for WHO based on existing evidence.[32] From the supply side, this framework depicted frontline health providers' attitudes and actions as paramount because they determine who receives services and under what conditions. Provider attitudes are influenced by their core beliefs, local values and norms, and empathy for clients. Their attitudes affect their actions, which can be modified through training and value clarification workshops, supervision, infrastructure, incentives, policies, and regulations. From the demand side, clients' use is influenced by their SRH knowledge, their felt need for service, the convenience of services, and costs (both financial and psychosocial).[32] To improve access to SRH services in rural African government health facilities, it is vital to conduct rigorous trials to test the impact of manipulating these supply-side and demand-side factors.

## Description of the intervention

Based on this conceptual framework, a Kenyan non-profit headquartered in Nairobi, the Centre for the Study of Adolescence (CSA)--with technical support from the Bixby Program at the University of California at Los Angeles (UCLA)-- conducted a randomized cluster study to test the efficacy of the After-Hours Adolescent Project (AHAP). After signing a memorandum of understanding in March 2018 with the Kenyan Ministry of Health, CSA/UCLA began collecting baseline data from thirteen government health facilities in Bungoma County, Kenya. In June 2018, the program started training recently-graduated nurses to be placed in the facilities. One year later, CSA/UCLA conducted an evaluation of AHAP and led a county-wide dissemination of the results.

AHAP aimed to increase adolescent clients' use of SRH services by building on existing resources available at rural government facilities. The two core elements were: (1) to keep health facilities open after hours, both to address the convenience factor and to reduce community members' knowledge of youths' use of services, and (2) to hire new young nurses to reduce the psychosocial costs associated with young people obtaining SRH services. The intervention sought to improve providers' attitudes and actions by training them on youth-friendly services (YFS) and comprehensive sexuality education (CSE), ensuring they had adequate supplies of condoms at the facilities, and instituting monthly supervisor visits to problem-solve any difficulties.

The AHAP intervention components were intended to influence specific aspects of the conceptual framework developed for WHO (see Fig 1). We hypothesized that adolescent use of SRH services would be strongly affected by providers' approachability and accessibility. At the same time, when more adolescents use the services, providers might be more inclined to see that young people have diverse needs and deserve individualized attention, which could in turn have a positive effect on their attitudes. Supply-side factors tested by AHAP focused on the age of providers, years of practice, training on YFS and CSE, and adequate infrastructure and supplies. Demand-side factors tested were convenience, privacy, better SRH knowledge through CSE, rapport with providers, and reduction of psychosocial costs to young people.

The study tested two versions of AHAP—the full intervention and a partial intervention. The full intervention differed from the partial intervention in that it included having the AHAP nurse conduct CSE sessions in schools and the community. We wished to test if adding this element increased young people's familiarity with the nurse and overcame some psychological barriers to access, as compared with the partial intervention which was entirely clinic-based. The full intervention was comprised of these five components: (1) convincing rural government health facilities to keep facilities open after normal hours (later on weekdays and for half-day on Saturdays); (2) recruiting one recent nursing school graduate to be posted in each participating health facility; (3) training the young nurses in YFS and CSE; (4) creating a cadre of rovers (details provided below) who provided logistics support to nurses conducting CSE in the community; and (5) improving facility infrastructure by adding lighting (if necessary), board games and informational brochures, and ensuring adequate availability of condoms. While the partial intervention did not include nurse-led CSE in the community, it did encompass all the other components.

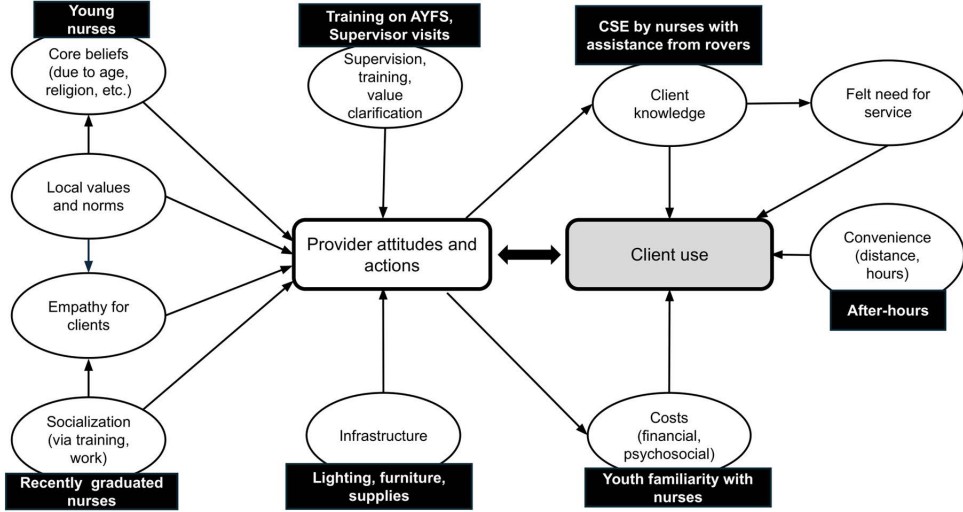

**Fig 1. Components of AHAP (black boxes), based on conceptual framework developed for WHO [32].**

In the full intervention, the nurses sometimes conducted CSE in the late morning and arrived in the afternoon at the clinics, remaining there until dark. Each nurse visited 4–5 schools in their catchment area, which included primary and secondary schools for both boys and girls. The nurses were assisted by a new cadre of rovers set up by the program. Rovers were young high school graduates who were in their waiting period between completing their high school examinations and possibly joining a college. These rovers helped nurses with the logistics of setting up the CSE classes at schools to make the process more efficient. They also helped the AHAP nurses occasionally conduct CSE on the weekend or in community centers to reach out-of-school youth, young couples, and parents. In contrast, for the partial intervention, nurses worked a staggered schedule in the facilities (generally from 10:30 AM to 6:30 PM) and did no community outreach. In both intervention types, AHAP nurses also were to work one half-day on the weekend. Special clinic registers were provided to record visits that took place during the extended hours.

Ten new nursing graduates (23–29 years old) were recruited specifically for the provision of SRH services to adolescents at the intervention facilities. Nurses underwent a 3-day training with role plays, values clarification, and discussions on adolescent-friendly services to ensure their alignment with the program. AHAP trained nurses on the importance of ensuring equitable, accessible, acceptable, appropriate, and effective care. They also received a 5-day training on how to provide adolescents with CSE and dispel SRH myths. The CSE curriculum included information on puberty, abstinence, consent, contraception, sexually-transmitted infections, peer pressure, sexual harassment and rape, and young people's rights to SRH services. None of the nurses had previously received CSE training.

Lastly, AHAP carried out basic facility-level infrastructure and supply improvements. Given low electricity availability, solar panels were installed at some facilities so that services could be provided during twilight and even after dark. The facilities were also made more attractive to youths by setting up basic adolescent corners with board games and information brochures on SRH. Chairs and tables were arranged to provide privacy during consultation, and supply cabinets for condoms and registers were purchased. Anonymous client cards and lockboxes were also provided so that youth clients could give their confidential feedback on services received.

## Methods

The purpose of the study was to test whether making SRH services more convenient, private, confidential, and youth-friendly increased young people's use of rural government health facilities. The team approached government health

facilities in a rural county of Western Kenya to invite them to participate in the study. Of the fifteen facilities approached, thirteen agreed to participate. Through a randomization process, five facilities were assigned to receive the full intervention, four were assigned to the partial intervention, and four were not assigned to any intervention (control facilities).

The study was conducted between April 2018 and June 2019. Baseline data were collected from all facility registers for the month of April 2018 before the program began, and included information on clinic opening hours and number of young people visiting for SRH and non-SRH reasons. The AHAP team recruited recent nursing graduates to serve as AHAP nurses in May 2018. A baseline survey of their knowledge, attitudes, and perceptions of adolescent sexual health and services was collected prior to training. During the project evaluation, endline data was collected from facility registers for the month of April 2019, following the same parameters as the baseline. Nurses were also re-surveyed.

In addition, both the full and partial intervention facilities were provided with feedback cards for clients which allowed young people to share their satisfaction with the services provided and give suggestions for improvement. Youths put their cards in locked boxes which only the AHAP evaluation team could access. Feedback cards were distributed from July 2018 to June 2019. The cards asked for information on client age, gender, facility visited, reason for visit, and provider's gender. They also included questions about services such as privacy, availability of supplies, and quality of interaction with the provider. Lastly, two focus group discussions were conducted--with AHAP nurses and with youths in a facility catchment area--to capture their experiences with the project.

F-tests and generalized linear mixed models were used to compare groups as appropriate, with $p < 0.05$ considered statistically significant. F-tests were conducted on AHAP nurses' SRH attitudes and beliefs regarding youth's utilization of SRH services prior to the training and at one-year follow-up, to determine whether nurses' attitudinal changes were sustained. To assess if youth clients' demographics and satisfaction differed by intervention type, we used generalized linear mixed models. Each model was run separately for each factor, with the intervention type (full vs partial AHAP) as the outcome variable, to control for any facility effects (as clusters). The predictors of interest were: client age, client gender, provider gender, and waiting time. Data was analyzed using SPSS version 30.

Regarding ethical approval, the County Government of Bungoma – Department of Health provided written approval of the project prior to implementation. The UCLA Institutional Review Board determined that the AHAP study was exempted because it involved a review of clinic registers and program evaluation data of a local implementing organization. Additional information regarding the ethical, cultural, and scientific considerations specific to inclusivity in global research is included in the Supporting Information.

## Results

The key finding of this study was that facilities with AHAP nurses significantly increased the number of SRH service visits by young people, with the greatest change occurring in the full intervention facilities. At baseline, the thirteen facilities were serving a catchment of nearly 105,000 people of all ages, with 144 clinic staff (see Table 1). Monthly youth SRH visits in the AHAP facilities at baseline in April 2018 totaled 1,031, which rose to 1,931 in April 2019 after ten months of the project, representing an 87.3% increase. Meanwhile, the control facilities as a whole had 337 monthly youth SRH visits at baseline and 337 at endline, which indicated that there had been no change. The full intervention facilities registered overall a 96.8% visit increase (even though one of the facilities lost their AHAP provider in the month before the endline assessment), which was significantly higher than the 77.2% increase in the partial intervention facilities.

In addition, the study found that 52.2% of the SRH visits in April 2019 occurred after hours in the full intervention facilities, and 37.6% in the partial intervention facilities, which suggested that the increase in youth SRH visits arose mainly during times when facilities generally were closed. The registers also showed that AHAP nurses usually kept the facilities open until 6 or 7 PM on weekdays, although one facility closed at 5 PM and one hospital stayed open until 8 PM. Moreover, most facilities did offer some Saturday hours, although two never opened on the weekends.

**Table 1.** SRH visits by adolescents in study facilities, before and during intervention period.

| Facility | Catchment population | No. of staff | No. of SRH youth visits (April 2018) | No. of SRH youth visits (April 2019) | Change over time (%) | Came after hours (%) | Weekday hours of AHAP nurse | Saturday hours of AHAP nurse |
|---|---|---|---|---|---|---|---|---|
| **Full AHAP** | | | | | | | | |
| Facility F1* | 18,177 | 23 | 272 | 254 | −6.6 | 57.3 | 11−7 PM | 12−6 PM |
| Facility F2 | 4,378 | 3 | 13 | 122 | 838.5 | 41.4 | 10−6 PM | None |
| Facility F3 | 4,614 | 2 | 47 | 201 | 327.7 | 71.2 | 10−6 PM | 10−6 PM |
| Facility F4 | 4,903 | 4 | 163 | 299 | 83.4 | 66.2 | 11−6 PM | 11−6 PM |
| Facility F5 | 11,585 | 7 | 37 | 171 | 362.2 | 24.9 | 9−7 PM | 9 −7 PM |
| **Subtotal** | **43,657** | **39** | **532** | **1047** | **96.8** | **52.2** | | |
| **Partial AHAP** | | | | | | | | |
| Facility P1 | 2,722 | 6 | 51 | 160 | 213.7 | 56.1 | 10−10 PM | 8-10 AM |
| Facility P2 | 5,938 | 2 | 91 | 146 | 60.4 | 17.4 | 9 −5 PM | 10−4 PM |
| Facility P3 | 2,412 | 3 | 122 | 219 | 79.5 | 19.5 | 9:30–6 PM | None |
| Facility P4 | 26,239 | 75 | 235 | 359 | 52.8 | 57.4 | 11− 8 PM | 11−8 PM |
| **Subtotal** | **37,311** | **86** | **499** | **884** | **77.2** | **37.6** | | |
| **Control** | | | | | | | | |
| Facility C1 | 5,172 | 4 | 65 | 87 | 33.8 | --- | None | None |
| Facility C2 | 3,783 | 3 | 47 | 49 | 4.3 | --- | None | None |
| Facility C3 | 5,118 | 3 | 74 | 53 | −28.4 | --- | None | None |
| Facility C4 | 9,958 | 9 | 151 | 148 | −2.0 | --- | None | None |
| **Subtotal** | **24,031** | **19** | **337** | **337** | **0** | | | |

* Note: AHAP nurse left in early March 2019 prior to the endline assessment in April 2019. The nurse was not replaced.

The study also conducted an assessment of nurse attitudes at baseline and endline on family planning methods, sexual norms, and HIV/AIDS issues with answers based on a 5-point scale: (1) Strongly disagree; (2) Disagree; (3) Unsure; (4) Agree; (5) Strongly agree. Of the ten nurses who were trained at baseline, seven took the survey prior to the training. At follow-up, these seven nurses and one new nurse (who had replaced an AHAP nurse midway through the project) took the survey. Of the twelve attitudes assessed, the study found that nurse attitudes shifted significantly in a more "youth-friendly" direction for six of the attitudes, of which most related to family planning (see Table 2). The remaining six attitudes also shifted in the desired direction, but the change was not significant. Notably, none of the nurses still believed that contraceptives were harmful to adolescents, that condoms had holes in them, or that young people needed parental permission to obtain family planning.

Client feedback cards were also assessed by the study team. A total of 1954 cards were retrieved from locked boxes in AHAP facilities, of which 1923 included the gender of the youth and were used for analysis (See Table 3). No youth reported being transgender. Female youths were more likely than male youths to fill out cards (55% to 45%). Nearly three-fourths of the youths were 21 or younger. For both genders, about 44% wished to be tested for HIV/STIs. Male clients were more likely to request condoms (42% vs 2%), while female clients were more likely to request some form of hormonal family planning method (47% vs. 2%). Nearly 12% of female clients desired a pregnancy test, post-abortion care, or antenatal/postnatal care. About 8% of all youths reported that they came for counseling only. Most of the care offered was from male AHAP nurses (71%).

Overall, client satisfaction seemed to be very high, with 96–99% of the youths who filled in the cards reporting that they received the information and services they wanted, were comfortable coming for services, had enough privacy, and the nurse was not harsh to them. There were no differences reported by gender. Waiting times overall were very low, with 92% of the clients reporting wait times of less than 15 minutes. The median wait time was five minutes.

**Table 2. Average scores on attitudes and beliefs of AHAP nurses, at baseline and follow-up (12 months later)\*.**

| Statement | Baseline (n=7) | Follow-up (n=8) | F | p-value |
|---|---|---|---|---|
| **Family Planning** | | | | |
| Some family planning methods cause cancer. [b] | 2.43 | 1.25 | 6.01 | **0.029** |
| If a girl uses family planning methods for long, she can become infertile. [b] | 2.57 | 1.38 | 5.99 | **0.029** |
| People who are 13 years of age are mature enough to make decisions regarding sexual activity. [a] | 1.57 | 4.00 | 37.11 | **0.000** |
| All people should have the right to choose what happens to their own body. [a] | 3.71 | 4.88 | 6.35 | **0.026** |
| If an adolescent is requesting family planning, parents should be notified first. [b] | 2.14 | 1.13 | 4.29 | 0.059 |
| **Sexual Norms** | | | | |
| Someone who has several sexual partners is a slut. [b] | 3.14 | 1.38 | 14.13 | **0.002** |
| When a girl had been sexually assaulted, she usually had provoked it. [b] | 1.43 | 1.25 | 0.30 | 0.595 |
| Young people should abstain from sex before marriage. [b] | 4.00 | 2.50 | 6.07 | **0.029** |
| About half of rape accusations are false. [b] | 2.43 | 2.00 | 0.76 | 0.399 |
| **HIV/AIDS** | | | | |
| It is not a guarantee that a person will get HIV if he/she has unprotected sex with an HIV positive person. [a] | 2.86 | 3.63 | 1.07 | 0.320 |
| Some condoms have pores in them. [b] | 2.43 | 1.50 | 2.66 | 0.127 |
| HIV can be transmitted through kissing. [b] | 2.14 | 1.38 | 2.25 | 0.158 |

\*Notes: The data presented are average scores using a 5-point scale: (1) Strongly disagree; (2) Disagree; (3) Unsure; (4) Agree; (5) Strongly agree. Scores can range from 1 to 5. Significant p-values are in bold. [a]For these items, more agreement (a higher score) represented the desired attitude. [b]For these items, more disagreement (a lower score) represented the desired attitude.

Using generalized linear mixed modelling to account for facility cluster effects, we compared the full AHAP intervention to the partial intervention to determine if there were any significant differences in patient demographics or waiting time, after controlling for the gender of the provider (see Table 4). The analysis of the client cards suggests that younger youths aged 10–17 were significantly more likely to be obtaining SRH services in the full AHAP facilities (aOR: 3.15, p=.003). In addition, it seems that female youths were significantly more likely to obtain services at full intervention facilities than males (aOR: 2.00, p=.008). Waiting times also appeared to be lower at the full intervention facilities.

In a focus group, adolescents said that they considered AHAP nurses to be very friendly. They found it empowering to be able to ask for an AHAP nurse by name when they visited the facility for SRH services. If the AHAP nurse was not present, adolescents reported leaving without being seen to avoid being embarrassed by other health providers. They appreciated the educative and nonjudgmental approach of AHAP nurses, as well as the privacy and confidentiality of the AHAP services in the late afternoon when the facilities had no other clients.

AHAP nurses in their focus group described that they liked how AHAP brought together schools and health facilities through the CSE instruction in the full intervention. They felt privacy and trust were the most important issues for young people, and providing CSE in the community helped adolescents overcome their shyness. The nurses stressed that "information is power" and CSE helped young people make sound decisions. Adolescents told them that their teachers were biased, and they appreciated having a nurse to whom they could talk openly about their SRH concerns. The nurses also noted that some school principals at elementary schools were pleased that during AHAP no pregnancies among school

**Table 3. Youth client services use and satisfaction in AHAP facilities, by gender of client, from client satisfaction cards (N = 1923).**

| | Male clients N = 866 | Female clients N = 1057 | Total clients N = 1923 | Chi-square | p-value |
|---|---|---|---|---|---|
| **Intervention type** | | | | 29.16 | **<.001** |
| Partial AHAP | 465 (53.7%) | 437 (41.3%) | 902 (46.9%) | | |
| Full AHAP | 401 (46.3%) | 620 (58.7%) | 1021 (53.1%) | | |
| **Age (years)** | | | | 23.06 | **<.001** |
| 10-17 | 168 (19.6%) | 305 (29.1%) | 473 (24.8%) | | |
| 18-21 | 443 (51.6%) | 487 (46.4%) | 930 (48.8%) | | |
| 22-30 | 247 (28.8%) | 257 (24.5%) | 504 (26.4%) | | |
| **Main service received** | | | | 863.71 | **<.001** |
| VCT/STI test | 383 (44.2%) | 279 (26.4%) | 662 (34.4%) | | |
| Condoms | 362 (41.8%) | 25 (2.4%) | 387 (20.1%) | | |
| Family planning | 16 (1.8%) | 339 (32.1%) | 355 (18.5%) | | |
| Family planning + VCT | 7 (0.8%) | 155 (14.7%) | 162 (8.4%) | | |
| Counseling only | 74 (8.5%) | 83 (7.9%) | 157 (8.2%) | | |
| Preg. test, ANC/PNC, or post-abortion care | 0 (0.0%) | 125 (11.8%) | 125 (6.5%) | | |
| Other | 10 (1.2%) | 20 (1.9%) | 30 (1.6%) | | |
| No response | 14 (1.6%) | 31 (2.9%) | 45 (2.3%) | | |
| **Gender of provider** | | | | 1.89 | .169 |
| Male nurse | 602 (69.5%) | 765 (72.4%) | 1367 (71.1%) | | |
| Female nurse | 264 (30.5%) | 292 (27.6%) | 556 (28.9%) | | |
| **Reported waiting time** | | | | 6.44 | **.040** |
| 0–5 mins. | 453 (58.5%) | 510 (54.5%) | 963 (56.3%) | | |
| 6–15 mins. | 278 (35.9%) | 345 (36.9%) | 623 (36.4%) | | |
| 16+ mins. | 44 (5.7%) | 81 (8.7%) | 125 (7.3%) | | |
| **Satisfaction with services** | | | | | |
| Comfort with provider | 839 (99.3%) | 1045 (99.7%) | 1884 (99.5%) | 2.28 | .320 |
| Obtained info. wanted | 845 (100.0%) | 1042 (99.6%) | 1887 (99.8%) | 3.24 | .072 |
| Obtained tests or supplies wanted* | 837 (99.4%) | 1034 (99.1%) | 1871 (99.3%) | 0.50 | .780 |
| Provider was not harsh | 815 (96.0%) | 996 (96.6%) | 1811 (96.3%) | 1.09 | .581 |
| Had enough privacy | 827 (98.5%) | 1027 (98.8%) | 1854 (98.7%) | 1.52 | .469 |
| Would be comfortable coming again | 835 (99.1%) | 1031 (99.4%) | 1866 (99.3%) | 0.86 | .649 |

Notes: *Includes those who didn't want anything. Abbreviations: Preg. = pregnancy; ANC/PNC = antenatal care/postnatal care; VCT/STI = voluntary counselling & testing/sexually-transmitted infections.

https://doi.org/0.1371/journal.pone.0347329.t003

girls had occurred, because normally several girls in grades 6–8 dropped out each year due to unwanted pregnancies. The nurses were also able to effectively deal with any community concerns about the program. During a dissemination workshop in the county that included AHAP nurses, one facility in-charge explained:

*Youths themselves are very particular. If it is an AHAP nurse, they come. If it is someone else, they won't come. At first, we had some challenges because so many young people were coming after hours. One school principal wanted to know why students are there [at the facility] at 5-6 PM. We held a stakeholders meeting and sensitized the community [about AHAP]. After that, we had no problems.*

**Table 4. Comparison of full AHAP to partial AHAP intervention, by client satisfaction cards, using generalized linear mixed modelling.**

| Variable | Total Client Cards N | Cards from Full AHAP facilities N (%) | Unadjusted Odds Ratio (OR) | Adjusted Odds Ratio (aOR) | p-value |
|---|---|---|---|---|---|
| **Age** | | | | | |
| 10-17 | 473 | 310 (65.5%) | 2.62 (2.02-3.39) | 3.15 (1.55-6.40) | **.003** |
| 18-21 | 930 | 496 (53.3%) | 1.66 (1.32-2.09) | 1.80 (0.95-3.41) | .070 |
| 22-30 | 504 | 212 (42.1%) | 1 | 1 | |
| **Client gender** | | | | | |
| Female | 1057 | 620 (58.7%) | 1.65 (1.37-1.97) | 2.00 (1.22-3.27) | **.008** |
| Male | 866 | 401 (46.3%) | 1 | 1 | |
| **Provider gender** | | | | | |
| Female nurse | 556 | 371 (66.7%) | 2.21 (1.80-2.71) | 1.71 (.987-2.30) | .055 |
| Male nurse | 1367 | 650 (47.5%) | 1 | 1 | |
| **Waiting time** | | | | | |
| 0–5 mins. | 963 | 488 (50.7%) | 1 | 1 | |
| 6–15 mins. | 623 | 440 (70.6%) | .131 (.084−.205) | .121 (.040−.365) | **<.001** |
| 16 + mins. | 125 | 30 (24.0%) | .307 (.200−.472) | .273 (.095−.788) | **.018** |

Notes: Adjusted Odds Ratio (aOR) included the full model with age, client gender, provider gender, and waiting time, while controlling for facility. P-values were calculated using the Wald test for generalized mixed linear models.

## Discussion

This study used a randomized cluster design to evaluate the effectiveness of a system improvement intervention to increase adolescent uptake of SRH services at rural government health facilities in Kenya. The two core strategies were to extend the hours of services into the early evening and on one day of the weekend, and to employ newly-graduated nurses who were trained in YFS and CSE. Our evaluation found that in facilities that implemented the full AHAP intervention, the number of young people utilizing SRH services almost doubled after less than one year (increase of 96.8%) as compared to no change at all in the control facilities. In facilities that implemented the partial AHAP (without the community engagement component), the number of young SRH clients still increased by 77%.

Extending facility hours appeared to be a critical component of the intervention, since half of the young people in the full intervention and one-third in the partial intervention accessed services outside of the normal operating hours. Other studies have noted that inflexible and inconvenient opening times of facilities are a major access barrier to adolescents' use of SRH services, with regular facility hours overlapping with school hours, resulting in risks to students' privacy by having to obtain their teacher's permission.[8,14] Additionally, excessive waiting times and frequent absences of staff discourage young people. Our findings align with previous research that had indicated that convenient facility hours improved adolescent use of HIV testing, family planning methods, and STI management in Africa.[33,34]

Another barrier that AHAP addressed was the dearth of SRH providers in rural facilities with whom young people felt comfortable consulting. Fear of judgment from older providers, who may be perceived as parental figures, makes it difficult for adolescents to communicate their needs.[19] Moreover, researchers have found that long-term socialization into existing health systems can result in reduction in empathy towards patients.[32] AHAP's hiring of recently-graduated young nurses meant that adolescents were meeting with nurses closer to their age who had not become inured to the problems faced by youths. Regarding adolescents' preferences for younger providers, previous research had been mixed. Some studies suggested that younger providers are preferred as they can better understand the SRH challenges of youth; however, other studies indicated a preference for older providers because they were perceived to be more competent or

experienced.[35–37] We hypothesize that AHAP's intensive training ensured that staff were competent even though they were young, which meant that they met both preferences identified in previous studies.

The AHAP training and implementation seemed to realign nurses' attitudes towards family planning services for young people, and other progressive norms. Even though the AHAP nurses were recent graduates, at the onset of the program they held some inaccurate and problematic beliefs about SRH care for adolescents, as seen in their baseline attitudinal scores. The AHAP training and their subsequent interactions with young people appeared to result in a sustained improvement in nurses' attitudes and beliefs, as evidenced by their scores at endline. More providers recognized that adolescents had the right to services without parental consent, and fewer believed negative myths about health effects of family planning methods or the unreliability of condoms. These results were in line with previous studies in Kenya, Malawi and Nigeria, which had found that capacity-building and supportive supervision of health providers improved their attitudes, such as recognizing their adolescents' rights to obtain contraceptives.[31,34,38,39]

In comparing the full to the partial intervention, we found significant differences in uptake of SRH services by the gender and age composition of the clients. The full intervention facilities seemed to attract significantly more female and younger clients. These results have parallels in previous studies, which had found that a combination of facility improvement and community outreach was more effective than single-component interventions.[9] Having school-based and community-based CSE sessions led by an AHAP nurse from a nearby facility may have overcome the anxieties of female and younger youths, who are more likely to be deterred from using SRH services due to worries about nurse shaming and parental punishment. Previous studies in Zambia and Kenya have also found that integrated CSE by trained providers can result in better SRH outcomes amongst adolescents, including reductions in adolescent pregnancy.[40,41] We hypothesize that a key factor influencing the effectiveness of the CSE component was the role of the rovers. This cadre allowed nurses not to have to spend extra time in the planning and logistics of offering CSE classes, making the process more efficient and feasible.

While the increase in adolescent uptake of SRH services was an intended outcome, our implementation also revealed some unintended benefits that the study did not measure. The program seemed to have had high levels of acceptability amongst multiple stakeholders. The addition of young, recently graduated nurses provided facility in-charges with more human resources. Although most facilities could not retain these staff post program because of budgetary constraints, during the course of the intervention period the AHAP nurses supported facility staff in their tasks beyond adolescent care and ensured that the facility stayed open. We also heard anecdotally that female drop-outs due to pregnancy from schools near AHAP facilities had reduced, resulting in parents and teachers being supportive of the program.

Our study had several limitations. First, it was conducted in only 13 facilities in one county in Western Kenya, so it may not be widely generalizable. However, the sample included a range of rural government facilities--from dispensaries and health centers to district hospitals--which suggests that AHAP could be effective across the spectrum of public facilities. Another limitation was that the study did not assess the difference in costs between the full and partial interventions. Even though the full intervention significantly increased uptake as compared to the partial intervention (97% vs 77%), it is possible that the added expense and complexity of the community-level CSE activities would not be feasible for some localities. Another limitation relates to the intervention itself. Although AHAP was highly acceptable and achieved dramatic improvements in service use, most facilities did not retain AHAP nurses after the conclusion of the program because they had not included their salaries in their annual budgets. It would be important to ensure at the onset that facilities would be prepared to absorb the cost of an additional salary at the conclusion of the program. Lastly, it is not known whether the intervention might have had more impact if it had lasted longer, as word-of-mouth about the services' convenience and youth-friendliness might have attracted growing numbers of young people.

Our study also had several strengths. First, our randomized cluster design added validity and rigor to our findings, which may not be the case with other similar studies, and responded to the call for more experimental studies on intervention effectiveness. Second, our evaluation used multiple methods to provide insights on the impact of the intervention

on the uptake of SRH services among adolescents, as well as on changes in types of services availed, client satisfaction, and attitudes of AHAP providers. This permitted a more comprehensive assessment of the intervention's effectiveness. Lastly, AHAP relied on existing government health facilities and schools, which meant that the program was grounded in the local context and responsive to local challenges.

Because AHAP was implemented prior to the COVID-19 pandemic, some may wonder if it would be effective post-pandemic. In its early stages, the pandemic significantly affected the delivery of health services--including SRH services for adolescents in Kenya--since many facilities either closed or offered only urgent care. Moreover, public schools were shuttered, which led to the rise of unintended pregnancies among schoolgirls in Kenya and other African countries.[42] During the pandemic, AHAP would not have been feasible because it relied on functioning schools and clinics. However, in the aftermath of the pandemic, schools and health facilities have largely returned to pre-pandemic operations, with very few ongoing service disruptions.[43] In this post-pandemic landscape, it is likely that AHAP would thrive.

Future research is needed to determine if AHAP would be feasible and have an equivalent impact on young people's use of SRH services if it were scaled up to the county or regional level. In settings where young people live equidistant from several government health facilities, it could also be valuable to experiment with a discrete choice approach to test whether young people would choose an AHAP facility over other facilities in a locality.

## Conclusion

This randomized cluster study demonstrated that it was possible, in less than a year, to nearly double young people's use of government health facilities for SRH services in rural Kenya. AHAP directly addressed some of the most entrenched psychosocial barriers facing young rural Kenyans. It allowed adolescents to obtain SRH services confidentially and privately, ensured that free condoms were always available, and put adolescents on more equal footing with young providers who were disinclined to be harsh or judgmental. Having AHAP nurses deliver CSE in schools and the community in the full intervention further helped to increase adolescents' familiarity with them and showed that they were not biased against young people. This combination of factors, not any single item, enabled a low-cost activity, relying entirely on government health facilities, to increase significantly adolescent SRH services use. Given the success of this small randomized cluster trial, we recommend that the Kenyan Ministry of Health scale up AHAP to an entire county or region to determine whether the approach is viable and produces similar results.

## Supporting information

**S1 Dataset. AHAP Youth Client Data for all facilities.**
(XLSX)

**S2 Dataset. AHAP Nurses Data for all facilities.**
(XLSX)

**S3 Dataset. AHAP Client cards for all facilities.**
(XLSX)

**S1 File. Inclusivity in global research.**
(DOCX)

## Author contributions

**Conceptualization:** Paula Tavrow, Albert Obbuyi.

**Data curation:** Paula Tavrow.

**Formal analysis:** Paula Tavrow, Krupa Maria Varghese.

**Investigation:** Paula Tavrow.

**Methodology:** Paula Tavrow.

**Project administration:** Albert Obbuyi.

**Resources:** Paula Tavrow, Albert Obbuyi.

**Supervision:** Paula Tavrow, Collins Juma.

**Validation:** Paula Tavrow.

**Writing – original draft:** Paula Tavrow, Krupa Maria Varghese.

**Writing – review & editing:** Paula Tavrow, Krupa Maria Varghese, Albert Obbuyi, Collins Juma.

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
