## [Decision Letter · Decision Letter 0]

4 Sep 2025

PONE-D-25-35983Increasing young people’s use of sexual and reproductive health services in government health facilities in rural KenyaPLOS ONE

Dear Dr. Tavrow,

Thank you for submitting your manuscript to PLOS ONE. After careful consideration, we feel that it has merit but does not fully meet PLOS ONE’s publication criteria as it currently stands. Therefore, we invite you to submit a revised version of the manuscript that addresses the points raised during the review process.

If applicable, we recommend that you deposit your laboratory protocols in protocols.io to enhance the reproducibility of your results. Protocols.io assigns your protocol its own identifier (DOI) so that it can be cited independently in the future. For instructions see: https://journals.plos.org/plosone/s/submission-guidelines#loc-laboratory-protocols. Additionally, PLOS ONE offers an option for publishing peer-reviewed Lab Protocol articles, which describe protocols hosted on protocols.io. Read more information on sharing protocols at . Additionally, PLOS ONE offers an option for publishing peer-reviewed Lab Protocol articles, which describe protocols hosted on protocols.io. Read more information on sharing protocols at https://plos.org/protocols?utm_medium=editorial-email&utm_source=authorletters&utm_campaign=protocols..

We look forward to receiving your revised manuscript.

Kind regards,

Alfredo Luis Fort, M.D., M.Sc., Ph.D.

Academic Editor

PLOS ONE

Journal Requirements:

https://journals.plos.org/plosone/s/file?id=ba62/PLOSOne_formatting_sample_title_authors_affiliations.pdf..

2. Please include a complete copy of PLOS’ questionnaire on inclusivity in global research in your revised manuscript. Our policy for research in this area aims to improve transparency in the reporting of research performed outside of researchers’ own country or community. The policy applies to researchers who have travelled to a different country to conduct research, research with Indigenous populations or their lands, and research on cultural artefacts. The questionnaire can also be requested at the journal’s discretion for any other submissions, even if these conditions are not met.  Please find more information on the policy and a link to download a blank copy of the questionnaire here: https://journals.plos.org/plosone/s/best-practices-in-research-reporting. Please upload a completed version of your questionnaire as Supporting Information when you resubmit your manuscript.”

3. During your revisions, please note that a simple title correction is required: Please change ""a randomized controlled trial"" to ""a cluster randomized controlled trial"". Please ensure this is updated in the manuscript file and the online submission information.

“Lalor Foundation and Save the Children (Sweden) supported the program implementation, which was in this study. Both funders had no role of any kind in study design, data collection, analysis, etc., and did not specifically fund the evaluation.”

“We are grateful for funding provided by the Lalor Foundation and Save the Children (Sweden). “

“Lalor Foundation and Save the Children (Sweden) supported the program implementation, which was in this study. Both funders had no role of any kind in study design, data collection, analysis, etc., and did not specifically fund the evaluation.”

7. Please remove all personal information, ensure that the data shared are in accordance with participant consent, and re-upload a fully anonymized data set.

Additional guidance on preparing raw data for publication can be found in our Data Policy (https://journals.plos.org/plosone/s/data-availability#loc-human-research-participant-data-and-other-sensitive-data) and in the following article: http://www.bmj.com/content/340/bmj.c181.long..

8. Please remove your figures from within your manuscript file, leaving only the individual TIFF/EPS image files, uploaded separately. These will be automatically included in the reviewers’ PDF.

10.If the reviewer comments include a recommendation to cite specific previously published works, please review and evaluate these publications to determine whether they are relevant and should be cited. There is no requirement to cite these works unless the editor has indicated otherwise.

Additional Editor Comments (if provided):

Thanks for the important study on an issue that is currently of key need to publish and be most aware for adolescents and youth. There are a few suggestions to improve the manuscript. See what you can do so that it is more readable and consistent with scientific expectations. Please also find a few recommendations in the attached file as well.

Reviewers' comments:

Reviewer's Responses to Questions

**Comments to the Author**

1. Is the manuscript technically sound, and do the data support the conclusions?

Reviewer #1: Partly

Reviewer #2: No

2. Has the statistical analysis been performed appropriately and rigorously? 

Reviewer #1: Yes

Reviewer #2: No

3. Have the authors made all data underlying the findings in their manuscript fully available?

Reviewer #1: Yes

Reviewer #2: No

4. Is the manuscript presented in an intelligible fashion and written in standard English?

Reviewer #1: Yes

Reviewer #2: Yes

5. Review Comments to the Author

Reviewer #1: The article provides a compelling case for improving SRH access among rural youth, and the AHAP model is clearly impactful. However, the data (2018–2019) predates COVID-19, which likely altered youth behaviour and service access patterns significantly.

It would strengthen the article to include a discussion or reflection on how the pandemic may have influenced SRH service delivery and adolescent needs in rural Kenya.

Consider exploring whether AHAP or similar interventions were adapted or sustained during the pandemic, and what lessons were learned.

The inclusion of more recent data or a follow-up study would greatly enhance the relevance and applicability of your findings in the current context.

Reviewer #2: Line 49-50 provide a citation

In the methods section, the analysis section needs more details to move beyond descriptive statistics

The analysis is too descriptive given the study design. How did you account for the clustering of observations within groups (clusters), acknowledging that individuals within the same cluster tend to be more similar? Consider more complex models for example Multilevel (Hierarchical) Models (Mixed-Effects Models)

You could also provide details about future research directions. One approach that could be used is to use Discrete Choice Experiments (DCEs) to elicit participants' preferences and use these to help design interventions to help improve service uptake among the adolescents.

6. PLOS authors have the option to publish the peer review history of their article (what does this mean?). If published, this will include your full peer review and any attached files.). If published, this will include your full peer review and any attached files.

.

Reviewer #1: **Yes:**Dr. Bin LiDr. Bin Li

Reviewer #2: **Yes:**Galven MaringwaGalven Maringwa

While revising your submission, please upload your figure files to the Preflight Analysis and Conversion Engine (PACE) digital diagnostic tool, https://pacev2.apexcovantage.com/. PACE helps ensure that figures meet PLOS requirements. To use PACE, you must first register as a user. Registration is free. Then, login and navigate to the UPLOAD tab, where you will find detailed instructions on how to use the tool. If you encounter any issues or have any questions when using PACE, please email PLOS at . PACE helps ensure that figures meet PLOS requirements. To use PACE, you must first register as a user. Registration is free. Then, login and navigate to the UPLOAD tab, where you will find detailed instructions on how to use the tool. If you encounter any issues or have any questions when using PACE, please email PLOS at figures@plos.org. Please note that Supporting Information files do not need this step.. Please note that Supporting Information files do not need this step.

---

## [Author Response · Author response to Decision Letter 1]

4 Nov 2025

Nov. 3, 2025

Thank you and the reviewers for your detailed and thoughtful comments. We have incorporated your feedback into our revised draft. Please find below our responses (in bold) to each point raised by the editor and reviewers.

https://journals.plos.org/plosone/s/file?id=wjVg/PLOSOne_formatting_sample_main_body.pdf a nd

https://journals.plos.org/plosone/s/file?id=ba62/PLOSOne_formatting_sample_title_authors_affili ations.pdf. - We have made the adjustments for file naming and followed the style

requirements as per the templates provided.

2. Please include a complete copy of PLOS’ questionnaire on inclusivity in global research in your revised manuscript. Our policy for research in this area aims to improve transparency in the reporting of research performed outside of researchers’ own country or community. The policy applies to researchers who have travelled to a different country to conduct research, research with Indigenous populations or their lands, and research on cultural artefacts. The questionnaire can also be requested at the journal’s discretion for any other submissions, even if these conditions are not met. Please find more information on the policy and a link to download a blank copy of the questionnaire here: https://journals.plos.org/plosone/s/best- practices-in-research-reporting. Please upload a completed version of your questionnaire as Supporting Information when you resubmit your manuscript.” - We have uploaded the

completed questionnaire and included more information about ethical approval in Kenya.

3. During your revisions, please note that a simple title correction is required: Please change "a randomized controlled trial" to "a cluster randomized controlled trial". Please ensure this is updated in the manuscript file and the online submission information. - We are no longer

putting the type of trial into the title of the manuscript. We have changed it to "a cluster randomized controlled trial" in the text.

“Lalor Foundation and Save the Children (Sweden) supported the program implementation, which was in this study. Both funders had no role of any kind in study design, data collection, analysis, etc., and did not specifically fund the evaluation.”

Please provide an amended statement that declares *all* the funding or sources of support (whether external or internal to your organization) received during this study, as detailed online

in our guide for authors at http://journals.plos.org/plosone/s/submit-now. Please also include the statement “There was no additional external funding received for this study.” in your updated Funding Statement.

Please include your amended Funding Statement within your cover letter. We will change the online submission form on your behalf. - We have now clarified that the study was paid for by the Lalor Foundation, and Save the Children covered the project implementation as part of a larger grant to CSA. We have also added, "There was no additional external funding received for this study" into the updated Funding statement and changed the Funding Statement in the cover letter.

When you resubmit, please ensure that you provide the correct grant numbers for the awards you received for your study in the ‘Funding Information’ section. - We have now matched the ‘Funding Information’ and ‘Financial Disclosure’ sections. We have a grant number for Lalor Foundation but not for Save the Children. The grant number for Lalor Foundation has been provided.

“We are grateful for funding provided by the Lalor Foundation and Save the Children (Sweden).“

“Lalor Foundation and Save the Children (Sweden) supported the program implementation, which was in this study. Both funders had no role of any kind in study design, data collection, analysis, etc., and did not specifically fund the evaluation.”

Please include your amended statements within your cover letter; we will change the online submission form on your behalf. - We have now removed the funding from the

Acknowledgements section and anywhere in the manuscript. We have only added

funding information in the Funding Statement and have added the funding statement in our cover letter.

7. Please remove all personal information, ensure that the data shared are in accordance with participant consent, and re-upload a fully anonymized data set.

Additional guidance on preparing raw data for publication can be found in our Data Policy (https://journals.plos.org/plosone/s/data-availability#loc-human-research-participant-data-and- other-sensitive-data) and in the following article: http://www.bmj.com/content/340/bmj.c181.long.

- We have now ensured that there are no hidden columns and fully anonymized the data set. We have reviewed the additional guidance shared and changed specific health

service dates to MM/YYYY format.

8. Please remove your figures from within your manuscript file, leaving only the individual TIFF/EPS image files, uploaded separately. These will be automatically included in the reviewers’ PDF. - We have now removed the figure from the manuscript and included it only as a separate image file.

9. Please include captions for your Supporting Information files at the end of your manuscript, and update any in-text citations to match accordingly. Please see our Supporting Information guidelines for more information: http://journals.plos.org/plosone/s/supporting-information. - We have done so for the inclusivity questionnaire.

10. If the reviewer comments include a recommendation to cite specific previously published works, please review and evaluate these publications to determine whether they are relevant and should be cited. There is no requirement to cite these works unless the editor has indicated otherwise. - Thank you for this clarification. We did not see any specific citations that we were asked to use. We have provided citations where they flagged as missing.

Additional Editor Comments (if provided):

Thanks for the important study on an issue that is currently of key need to publish and be most aware for adolescents and youth. There are a few suggestions to improve the manuscript. See what you can do so that it is more readable and consistent with scientific expectations. Please also find a few recommendations in the attached file as well. - We are glad that the editor

considers this an important study that needs to be published. We appreciate that

recommendations in the attached file and have made the requested changes. We have also sought to make it more readable and rigorous.

Reviewers' comments:

Reviewer's Responses to Questions

Comments to the Author

5. Review Comments to the Author

Reviewer #1: The article provides a compelling case for improving SRH access among rural youth, and the AHAP model is clearly impactful. However, the data (2018–2019) predates COVID-19, which likely altered youth behaviour and service access patterns significantly. - Youth behavior in rural Kenya was indeed altered during the height of the epidemic in

2020-21, when schools were closed and may facilities had skeleton staffs. But starting in late 2021, the situation returned to how it was in 2018-19. Our Kenyan collaborators

confirm this. We have now added a paragraph about this issue in the paper.

It would strengthen the article to include a discussion or reflection on how the pandemic may have influenced SRH service delivery and adolescent needs in rural Kenya. - Please see the comment above. The pandemic has not had lasting effect on youth behavior and needs. There was a rise in unwanted adolescent pregnancies during the Covid period, but that seems to have subsided. We have added a citation to that effect.

Consider exploring whether AHAP or similar interventions were adapted or sustained during the pandemic, and what lessons were learned. - AHAP was not sustained during the pandemic. We have now clarified this in the paper.

The inclusion of more recent data or a follow-up study would greatly enhance the relevance and applicability of your findings in the current context. - Unfortunately, we don't have data that is more recent. However, rural Kenya changes very slowly over time, so we srongly believe that the data presented in this paper is still relevant.

Reviewer #2: Line 49-50 provide a citation- We have now provided a citation to line 49-50.

In the methods section, the analysis section needs more details to move beyond descriptive statistics - Our data on use of facilities by youth clients only permits descriptive statistics, but because of the concern of this reviewer we did do generalized linear mixed modelling to test if there was a cluster effect by facilities. We present this new data in the paper

and describe the technique.

The analysis is too descriptive given the study design. How did you account for the clustering of observations within groups (clusters), acknowledging that individuals within the same cluster tend to be more similar? Consider more complex models for example Multilevel (Hierarchical) Models (Mixed-Effects Models)- As mentioned above, we now are presenting generalized

linear mixed modelling to take into account cluster effects in the client satisfaction card data.

You could also provide details about future research directions. One approach that could be used is to use Discrete Choice Experiments (DCEs) to elicit participants' preferences and use these to help design interventions to help improve service uptake among the adolescents.-

Thank you for this suggestion. We have now added a section on future research directions and have included mention of Discrete Choice Experiments.

Thank you again for your consideration of our manuscript. We look forward to proceeding with next steps on this publication.

Sincerely,

Paula Tavrow, PhD

Adjunct Professor, Department of Community Health Sciences UCLA Fielding School of Public Health

ptavrow@ucla.edu

(310) 435-1444

(On behalf of: Dr. Paula Tavrow, Krupa Maria Varghese, Albert Obbuyi, and Collins Juma)

---

## [Decision Letter · Decision Letter 1]

8 Dec 2025

PONE-D-25-35983R1Increasing young people’s use of sexual and reproductive health services in government health facilities in rural KenyaPLOS One

Dear Dr. Tavrow,

Thank you for submitting your manuscript to PLOS ONE. After careful consideration, we feel that it has merit but does not fully meet PLOS ONE’s publication criteria as it currently stands. Therefore, we invite you to submit a revised version of the manuscript that addresses the points raised during the review process.

If applicable, we recommend that you deposit your laboratory protocols in protocols.io to enhance the reproducibility of your results. Protocols.io assigns your protocol its own identifier (DOI) so that it can be cited independently in the future. For instructions see: https://journals.plos.org/plosone/s/submission-guidelines#loc-laboratory-protocols. Additionally, PLOS ONE offers an option for publishing peer-reviewed Lab Protocol articles, which describe protocols hosted on protocols.io. Read more information on sharing protocols at . Additionally, PLOS ONE offers an option for publishing peer-reviewed Lab Protocol articles, which describe protocols hosted on protocols.io. Read more information on sharing protocols at https://plos.org/protocols?utm_medium=editorial-email&utm_source=authorletters&utm_campaign=protocols..

We look forward to receiving your revised manuscript.

Kind regards,

Jianhong Zhou

Staff Editor

PLOS One

Journal Requirements:

Reviewers' comments:

Reviewer's Responses to Questions

**Comments to the Author**

1. If the authors have adequately addressed your comments raised in a previous round of review and you feel that this manuscript is now acceptable for publication, you may indicate that here to bypass the “Comments to the Author” section, enter your conflict of interest statement in the “Confidential to Editor” section, and submit your "Accept" recommendation.

Reviewer #2: (No Response)

2. Is the manuscript technically sound, and do the data support the conclusions?

Reviewer #2: Yes

3. Has the statistical analysis been performed appropriately and rigorously? 

Reviewer #2: No

4. Have the authors made all data underlying the findings in their manuscript fully available?

Reviewer #2: Yes

5. Is the manuscript presented in an intelligible fashion and written in standard English?

Reviewer #2: Yes

6. Review Comments to the Author

Reviewer #2: (No Response)

7. PLOS authors have the option to publish the peer review history of their article (what does this mean?). If published, this will include your full peer review and any attached files.). If published, this will include your full peer review and any attached files.

.

Reviewer #2: **Yes:**Galven MaringwaGalven Maringwa

---

## [Author Response · Author response to Decision Letter 2]

7 Feb 2026

We have now made the suggested statistical changes to Table 4, using the Wald statistic and adjusted Odds Ratios, taking into account facility cluster effects.

---

## [Editor Report · Decision Letter 2]

1 Apr 2026

Increasing young people’s use of sexual and reproductive health services in government health facilities in rural Kenya

PONE-D-25-35983R2

Dear Dr. Paula Tavrow,

We’re pleased to inform you that your manuscript has been judged scientifically suitable for publication and will be formally accepted for publication once it meets all outstanding technical requirements.

An invoice will be generated when your article is formally accepted. Please note, if your institution has a publishing partnership with PLOS and your article meets the relevant criteria, all or part of your publication costs will be covered. Please make sure your user information is up-to-date by logging into Editorial Manager at Editorial Manager® and clicking the ‘Update My Information' link at the top of the page. For questions related to billing, please contact  and clicking the ‘Update My Information' link at the top of the page. For questions related to billing, please contact billing support..

Kind regards,

Habil Otanga, Ph.D

Academic Editor

PLOS One

Additional Editor Comments (optional):

I note that the Reviewer suggested statistical changes to Table 4 using the Wald Statistic and Odds Ratios which were incorporated in the current draft as suggested.

I consider the draft manuscript to be aligned with Reviewers' suggestions and edits.
---

## [Editor Report · Acceptance letter]

PONE-D-25-35983R2

PLOS One

Dear Dr. Tavrow,

I'm pleased to inform you that your manuscript has been deemed suitable for publication in PLOS One. Congratulations! Your manuscript is now being handed over to our production team.

Kind regards,

on behalf of

Dr. Habil Otanga

Academic Editor

PLOS One